# Mortality in Via Ferrata Emergencies in Austria from 2008 to 2018

**DOI:** 10.3390/ijerph17010103

**Published:** 2019-12-22

**Authors:** Mathias Ströhle, Matthias Haselbacher, Christopher Rugg, Alex Walpoth, Ricarda Konetschny, Peter Paal, Peter Mair

**Affiliations:** 1Department of General and Surgical Critical Care Medicine, Medical University of Innsbruck, Anichstrasse 35, 6020 Innsbruck, Austria; mathias.stroehle@tirol-kliniken.at; 2Department of Trauma Surgery, Medical University of Innsbruck, Anichstrasse 35, 6020 Innsbruck, Austria; matthias.haselbacher@tirol-kliniken.at; 3Department of Orthopedics and Trauma Surgery, Medical University of Innsbruck, Anichstrasse 35, 6020 Innsbruck, Austria; alex.walpoth@student.i-med.ac.at; 4Department of Anaesthesiology and Intensive Care Medicine, Clinical Centre Wels-Grieskirchen GmbH, Grieskirchner Strasse 42, 4600 Wels, Austria; r.konetschny@gmx.at; 5Department of Anaesthesiology and Intensive Care Medicine, Hospitallers Brothers Hospital, Paracelsus Medical University Salzburg, Kajetanerplatz 1, 5010 Salzburg, Austria; peter.paal@icloud.com; 6Department of Anesthesiology and Intensive Care Medicine, Medical University of Innsbruck, Anichstrasse 35, 6020 Innsbruck, Austria; peter.mair@tirol-kliniken.at

**Keywords:** via ferrata, cable, climbing, mortality, accident, emergency, rescue

## Abstract

Although the European Alps now have more than 1000 via ferratas, limited data exist on the actual incidence of fatal events in via ferratas and their causes. This retrospective study analysed data from a registry maintained by the Austrian Alpine Safety Board (*n* = 161,855, per 11 September 2019). Over a 10-year period from 1 November 2008 to 31 October 2018, all persons involved in a via ferrata-related emergency were included (*n* = 1684), of which 64% were male. Most emergencies were caused by blockage due to exhaustion and/or misjudgement of the climber’s own abilities. Consequently, more than half of all victims were evacuated uninjured. Only 62 (3.7%) via ferrata-related deaths occurred. Falling while climbing unsecured was the most common cause of death, and males had a 2.5-fold higher risk of dying in a via ferrata accident. The mortality rate was highest in technically easy-to-climb sections (Grade A, 13.2%/B, 4.9%), whereas the need to be rescued uninjured was highest in difficult routes (Grade D, 59.9%/E, 62.7%). Although accidents in via ferratas are common and require significant rescue resources, fatal accidents are rare. The correct use of appropriate equipment in technically easy-to-climb routes can prevent the majority of these fatalities.

## 1. Introduction

A via ferrata—which literally translates to “iron path”—is a mountainous climbing route fully and permanently equipped with facilities like ladders, steps, cables and anchors fixed to the rock. Difficulty levels range from easy access, with many steps and footholds, to athletic passages, access difficulties (e.g., climbing sequences) and exposed vertical or overhanging sections equipped only with cables. Therefore, via ferrata climbing is practiced by a broad cross-section of the population with various experience levels and age ranges from very old to very young. In addition to general protective gear—especially gloves, partially containing shock absorbers—protective equipment mainly consists of a harness, a helmet and a special via ferrata, heavy-duty lanyard as a connection between the harness and the steel cable in the wall. Via ferrata kits are commercially available.

Historically, regarding via ferrata climbing ropes, cables, ladders and other technical equipment were fixed to the rock and used to facilitate accessibility on a mountain route. Some of the earliest via ferratas were installed on the Grossglockner in Austria in 1869 and the Marmolada in Italy in 1903. During the First World War, many alpine routes along the Austrian–Italian front were equipped with iron steps and cables to facilitate army access to austere alpine areas. In the early 1930s, some alpine climbing routes (e.g., Brenta region, Italy) were improved by installing iron steps and cables. Then, toward the end of the last century, a boom produced more than 1000 via ferratas in the Alps, mainly in Austria and Italy, but also in France and Switzerland. 

In Austria, via ferrata climbing has gained increasing popularity over recent years. As a consequence, the number of accidents has also risen steeply. Analysis of national registry data from 2008 to 2018 revealed that the number of accidents and rescue missions in via ferratas in Austria has increased by more than 60% (see Appendix A).

Until now, peer-reviewed literature reporting on via ferrata accidents, injury patterns and mechanisms, as well as outcome, has been scarce. To our knowledge, only two studies from Yosemite National Park, CA, have been published to date [1,2]. Both concentrated on Half Dome, a summit reached by a 23 km to 26 km roundtrip hike followed by the final 146 m, which is equipped with cable handrails. The aim of this study was to analyse the mortality rate, mechanisms, injury patterns and rescue requirements of via ferrata accidents using the Austrian national database. 

## 2. Materials and Methods

This retrospective study was approved by the Ethics Committee of the Medical University of Innsbruck (AN4757 315/4.4) and registered with Clinical Trials (NCT03405467). Since 1 November 2005, the Austrian Alpine Police has collected anonymised data on all accidents, as well as all fatal medical incidents, occurring in the Austrian mountains. An emergency dispatch centre call triggers entry and storage in a registry managed by the Austrian Alpine Safety Board. From 1 November 2005 to 11 September 2019, 108,181 emergencies involving 161,855 victims of various mountain activities were entered. The Austrian Alpine Safety Board Registry is one of the largest for mountain emergencies. Cross-country skiing, skiing, ski touring, snowboarding, toboggan, canyoning, rafting, hiking, caving, mountain biking, paragliding and hang-gliding were the activities which were excluded. Climbing activity was excluded when ice climbing, sport climbing, bouldering or alpine climbing was performed. All accidents occurring while climbing a via ferrata between 1 November 2008 and 31 October 2018 were retrieved from the registry and included in this analysis (Appendix A). Persons involved in emergencies while ascending to or descending from a via ferrata were also enrolled. The terrain outside the via ferratas was graded by police officers using the UIAA (Union Internationale des Associations d’Alpinisme) scale for difficulty of rock climbing routes [3]. Via ferratas were graded according to the scale developed by the Austrian climbing guide author Kurt Schall [4]. This scale—from A to E—conveys the difficulty levels easy, moderately difficult, difficult, very difficult and extremely difficult (Appendix A). Data obtained for this study included sex, age, route difficulty, injury severity (uninjured, injured, dead), accident cause (medical emergency, defined as any medical emergency other than exhaustion, trauma or special causes like rock- or icefall), equipment used, type of rescue mission (air rescue, ground rescue, combined) and weather conditions (Appendix A). Differences in characteristics between victims were calculated using Chi-square tests. Descriptive statistics are presented as mean ± SD or count and percentage, as appropriate. Data were stored with Excel 2019 (Microsoft, Seattle, WA, USA) and processed with IBM SPSS 24.0 (IBM, Armonk, NY, USA).

## 3. Results

### 3.1. Emergency Characteristics

During the study period, 1231 accidents or medical emergencies were identified in association with via ferrata climbing. In total, 1684 people were involved. Of them, 1092 (64.8%) were male and 592 (35.2%) were female. The mean age was 39 (range: 3 to 82) years. In the group of 20–29-year-olds, females were 1.6-fold more often involved in via ferrata accidents. In older groups, men were more frequently involved (50–59 years: 1.4-fold, 60–69 years: 1.7-fold and 70–79 years: 2.3-fold). The mean level of altitude was 1522 ± 622 m. Injury severity was not affected by the altitude of the accident site. Rescue missions were almost equally performed by helicopter and terrestrial teams (Table 1). Only a few individuals (*n* = 127, 7.5%) were rescued in a combined ground and air rescue mission. In comparison to ground-based rescue, helicopter rescue was more frequently performed at higher altitudes (1722 ± 590 m vs. 1273 ± 645 m, respectively). Weather was mostly sunny or cloudy (*n* = 1331, 79.5%, see Table 1). Most accidents occurred during the summer months from June to September, predominantly on Saturdays and Sundays (Figure 1). The mean time of day of an emergency was 14:25 ± 3:01 h. Thirty-nine (2.3%) persons had to be rescued in darkness.

### 3.2. Causes of Emergencies in Via Ferratas and Their Consequences

The leading causes of emergencies in via ferratas were exhaustion (*n* = 713, 42.3%), falls (*n* = 384, 22.8%) and being blocked or lost uninjured (*n* = 294, 17.5%) (Table 2). Other causes were bad weather, injuries not caused by falling, medical emergencies, rockfall, lightning strike, material failure and icefall. Two anaphylactic reactions caused by wasp stings, one snakebite and three strokes occurred as medical emergencie, without fatalities. In total, 960 (57.0%) persons were uninjured but in need of rescue, while 489 (29%) mountaineers suffered injuries and 62 (3.7%) persons died. 

A secured fall into a via ferrata set was documented in 189 (11.2%) cases, resulting in 106 injured, 50 uninjured and five dead mountaineers. Unsecured falls when not using a via ferrata set were recorded in 69 (4.1%) cases, resulting in 21 injured, 19 uninjured and 26 dead mountaineers. Equipment failure or malfunction was the cause of 49 emergencies (2.9%), resulting in 17 injured patients and three fatalities. Compared to the overall population, not using proper via ferrata equipment was 1.5-fold more frequent in the 10–19-year-olds, whereas the very old (70–79 years) were twice as often involved in accidents with equipment malfunction (6.5% vs. 2.9%). The relative risk for equipment failure was almost equal in males (3.0%) and females (2.7%), but women had a 1.9-fold higher risk of making mistakes while belaying. Incidents caused by not using a via ferrata set or equipment malfunction were 1.2- and 2.5-fold higher in men, respectively. The non-usage of a via ferrata set was also 1.8-fold higher in Austrian-natives than in the most frequent group of tourists (Germans). An unknown degree of injury was documented in 28 secured and three unsecured falls. Regarding rescue efforts due to exhaustion or medical emergencies, the vast majority (*n* = 643, 91.1% and *n* = 50, 92.6%, respectively) occurred during the ascent. The causes of emergencies that most commonly led to no injuries were exhaustion (*n* = 525, 73.6%), being blocked (*n* = 265, 90.1%) and bad weather conditions preventing progress on the via ferrata (*n* = 95, 97.9%) (Table 2). 

### 3.3. Emergencies Leading to Death

Sixty-two (3.7%) accidents resulted in the death of the involved person. Females suffered 11 fatalities, while males suffered 51. The risk of dying was 2.5-fold higher for males than for females. The risk of suffering deadly injuries was higher in older individuals (60–69 years, 2.8-fold; 70–79 years, 3.5-fold) and lower in the young (10–19 years and 20–29 years, 0.45-fold). The most common cause of death was a fall (*n* = 47, 75.8%), followed by a medical emergency (*n* = 7, 11.3%). Two cardiac arrests and one stroke caused death indirectly by provoking a fall. One death was attributed in the registry to exhaustion and one to getting lost. Fatal accidents occurred predominantly in easy via ferratas (Grades A and B). The height of lethal falls was at least five metres with a mean of 97 ± 72 m. The incorrect use or non-use of safety equipment was causative in 41 (87.2%) fatalities. More than two-thirds (*n* = 24) of all deadly accidents associated with not using safety equipment occurred in Grades A and B via ferratas. Of all deaths in Grade A, 90% were caused by not using safety equipment, 72.7% in Grade B, 66.6% in Grade C, 21.1% in Grade D and 100% in Grade E. Do-it-yourself safety equipment (e.g., slings, ropes) led to lethal falls in five cases and torn harness anchor slings in four cases. Overriding fall protection resulted in a lethal fall into the harness in two cases. One person died because of incorrect rope-handling while rappelling. There was a slight reduction in risk for lethal accidents when the tour was guided (1:1.3). 

### 3.4. Via Ferrata Difficulty Level in Relation to Injury

Most accidents occurred in Grade C (*n* = 385, 22.9%) or Grade D (*n* = 426, 25.3%) via ferratas (Table 3). Although most accidents were not associated with injuries (Grade C n = 218, 56.6%, Grade D *n* = 255, 59.9%), rescue missions in difficult via ferratas (Grades C, D) were most often undertaken for uninjured victims (*n* = 362, 94.0% and *n* = 403, 94.6%, respectively) (Appendix A). The highest rate of injured patients was experienced in Grade B via ferratas (*n* = 76, 34.2%). The UIAA climbing difficulty grade was reported in 120 cases where the accident did not happen directly in the via ferrata (e.g., terrain entering or exiting the via ferrata) and was 3.14 ± 1.2, indicating a terrain normally in need for climbing experience. The majority of emergencies occurred during the ascent (*n* = 1317, 78.2%). Situations when traversing a rock face horizontally or rappelling more frequently led to injuries and fatalities (Appendix A). 

## 4. Discussion

One important finding of this study is a mortality rate of 3.7% in via ferrata emergencies. Mortality is typically found in victims falling on routes graded as being technically easy and when using insufficient or no safety equipment. The majority of rescue missions are necessary for uninjured or only mildly injured persons and are due to minor falls into safety equipment or being lost, blocked or exhausted. 

Concerning via ferrata accidents, no data from larger or nationwide registries have been reported so far. The registry of the Austrian Alpine Safety Board contains 79,551 emergencies with 117,456 involved persons for the period from 1 November 2008 to 31 October 2018 and is the largest known registry with data from alpine accidents. As no comparable hospital data were available, the climbers were graded only uninjured, injured or dead.

### 4.1. Emergency Characteristics

With a male to female ratio of 1.8:1, although still predominant, the male fraction is lower than in other mountaineering sports [5,6]. Via ferrata climbing is obviously practiced by a broader cross-section of the normal population, which also fits with the wide age range observed in via ferrata emergencies. Climbing was performed by the very old, as well as the very young, underlining the fact that with the proper choice of route, via ferrata climbing can be done by anyone. This is in line with data from the Yosemite National Park, where an age range from 8–70 years was observed [2]. It should be expected, however, that both age extremes climb mainly easier tours. Differentiating by age and gender showed that especially young men were prone to being involved in a via ferrata accident because of non-usage of safety equipment, whereas old men instead dealt with equipment malfunction. The most predominant female contribution to via ferrata emergencies were mistakes while belaying. The distribution of via ferrata emergencies according to month and day of the week, with a concentration on summer weekends, shows the seasonal, recreational character of via ferrata climbing. This also indicates the economic potential of this sport as a business and tourism market. On Half Dome in Yosemite, where summer weekend peaks are also observed, overcrowding has already led to restrictions in access permissions [2].

We observed a larger number of uninjured (57%) as compared to injured persons involved in via ferrata emergencies, which is comparable to the 59% nontraumatic search and rescue missions from Yosemite [2]. This was also described for other mountain activities. A nationwide retrospective study, which was also conducted in the Austrian Alps, showed that 35.9% of all people involved in a canyoning related accident were uninjured [7]. The provided data in this study show that general exhaustion, blockage or getting lost while elsewise staying well make a major contribution to this ratio. The majority of all persons involved in via ferrata emergencies were practicing their sport in C- or D-graded via ferratas (Table 3), while more than half of them were rescued uninjured. Here, underestimation of the intensity and level of difficulty of the via ferrata is the suspected cause of exhaustion and/or getting stuck. Also, in accordance with the associated cardiovascular stress, almost all exhausted persons were reported during the ascent. Presumably, this is also the reason why the majority of medical emergencies occurred while ascending. 

The few emergencies that took place while rappelling seem to be extreme situations where climbers tried to leave a via ferrata by unconventional means, leading to increased injury and mortality rates. As we believe that only few persons use via ferratas to actively descend, it is logical that the largest proportion of all emergencies occurred while ascending. When traversing a rockface, the rate of injury increased, and mortality rate was twice as high. We conclude that underestimation of potential risks in these situations results in predominantly unsecured climbing, as all fatalities that occurred when traversing a wall were not secured. 

Rockfall and bad weather conditions were causative for accidents during the ascent, but rarely during the descent. It is noteworthy that people who were struck by lightning were mostly descending (56%), followed by ascending (28%) and traversing (17%). We suspect that this is most probably due to the fact that climbers attempt to escape by making a descent when weather changes quickly and thunderstorms approach. Neither rockfall nor lightning resulted in death when climbing a via ferrata.

### 4.2. Emergencies Leading to Death

Compared to a nationwide mortality of 6.2 deaths per year in Austria, the data from Yosemite show a total number of deaths of *n* = 12 in a 10-year timeframe for a single summit [2]. The differentiation of causes leading to death in the history of the summit revealed that 10 of 31 traceable deaths were related to hiking (*n* = 5) and cable handrails (*n* = 5) [1].

The proportion of accidents leading to injuries or death was highest when performing A- and B-Grade via ferratas. There are two possible explanations for this finding. First, an underestimation of the situation and the environment—by the less and more experienced—might play a key role in routes graded as being easy. Second, persons performing A- and B-Grade via ferratas might be generally less experienced and therefore at greater risk of suffering injuries or lethal accidents. We were able to show that nearly all fatalities occurring in Grade A via ferratas were caused by the inappropriate use or non-use of safety equipment. This led to a 3.6-fold increased mortality rate in Grade A via ferratas as compared to overall mortality for via ferrata accidents. As the police do not document the level of experience of persons involved in a via ferrata accident, we were unable to state whether this was a major contributor to the increased mortality rate or not. Most fatal accidents were the result of a fall, followed by medical emergencies and material failure. Whereas the injury rate was only 1.1-fold higher in females, males had a 2.5-fold higher risk of dying when climbing a via ferrata, which is in line with other mountaineering sports [5]. We conclude that males are more willing to take greater risks when climbing via ferratas. 

### 4.3. Prevalence

The exact prevalence of via ferrata climbing in Austria is unknown. However, at 6.2 deaths per year, the risk of dying is low. This stands in comparison to an average annual death rate of 110 while hiking, 20 while rock climbing, five while mountain biking and 0.7 while canyoning in the Austrian Alps [7,8]. The annual hours of via ferrata climbing in Austria can be estimated from data from the German Alpine Club, the biggest mountain sports club worldwide [9]. Providing information on injuries and deaths during certain sport activities, the German Alpine Club is able to relate the number of affected members to its total number of members. The German Alpine Club has stated that one injury occurs for every 217,000 hours and one fatality occurs for every 10 million hours of via ferrata climbing performed. Applying this to our observed average of 187 injuries and 6.2 deaths per year, we can conclude that 40–62 million hours of via ferrata climbing are performed annually in Austria.

### 4.4. Limitations

It must be stated that the frequency of via ferrata climbing in the Austrian Alps, as calculated from the number of hours per year, is unknown. Only estimates based on data from the German Alpine Club are possible. As most via ferratas are freely accessible, it is impossible to calculate exact numbers without prospective studies of athlete behaviour. Therefore, the exact calculation of incidence and mortality rate per hour of performance is impossible. From other studies, it is known that a reasonable number of victims are not included in the data kept by the Austrian Alpine Safety Board [10]. This is because only emergencies that send a distress call to an emergency dispatch centre are documented. This leads to an overestimation of injured persons and might therefore produce a bias in favour of an increased risk for injury. In reality, we know from other studies that many of the victims are able to evacuate themselves and seek medical help on their own, often in peripheral hospitals or from general practitioners [10].

## 5. Conclusions

The mortality of via ferrata emergencies during the study period was 3.7%. More than half of all fatal accidents occurred while climbing not belayed and without appropriate safety equipment. The risk of suffering a deadly accident in a via ferrata was greatest in via ferratas graded as being easy (Grades A/B). Mortality in via ferrata accidents can be further reduced by the consistent use of appropriate safety equipment on technically easy routes.

## Figures and Tables

**Figure 1 ijerph-17-00103-f001:**
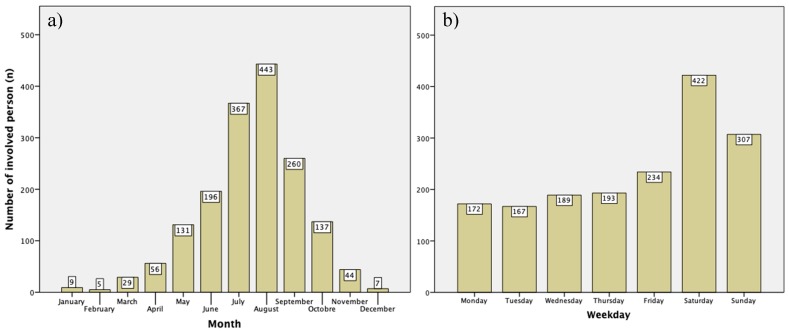
Months (**a**) and days of the week (**b**) of via ferrata emergencies in Austria from 1 November 2008 to 31 October 2018.

**Table 1 ijerph-17-00103-t001:** Characteristics of via ferrata accidents in Austria from 1 November 2008 to 31 October 2018. Listed according to frequency.

Characteristics		*n*	(%)
Rescue mission			
	Ground rescue	732	43.5
	Air rescue	725	43.1
	Combined ground and air rescue	127	7.5
	No rescue	89	5.3
	Unknown	11	0.7
Weather			
	Sunny	1067	63.4
	Cloudy	271	16.1
	Rainy	93	5.5
	Sudden fall in temperature	84	4.9
	Thunderstorm	57	3.4
	Fog	36	2.1
	Blizzard	6	0.4
	Unknown	70	4.1

Note: Unknown: Police officers were unable to obtain information.

**Table 2 ijerph-17-00103-t002:** Cause of emergency vs. injury classification on via ferratas in Austria from 1 November 2008 to 31 October 2018. Listed according to total frequency.

Cause of Emergency	Uninjured *n* (%)	Injured *n* (%)	Dead *n* (%)	Unknown *n* (%)	Total *n* (%)
Exhaustion	525 (73.6)	147 (20.7)	1 (0.1)	40 (5.6)	713 (42.3)
Fall	39 (10.2)	220 (57.5)	47 (12.2)	77 (20.1)	384 (22.8)
Blocked or lost person	265 (90.1)	21 (7.1)	1 (0.3)	7 (2.4)	294 (17.5)
Bad weather	95 (97.9)	2 (2.1)	0	0	97 (5.8)
Injury not caused by fall	9 (13.0)	36 (52.2)	0	24 (34.8)	69 (4.1)
Medical emergency	19 (34.5)	21 (38.3)	7 (12.7)	8 (14.5)	55 (3.3)
Rockfall	0	30 (81.1)	0	7 (18.9)	37 (2.2)
Lightning strike	1 (5.6)	7 (44.4)	0	10 (55.6)	18 (1.1)
Material failure	0	2 (33.3)	4 (66.7)	0	6 (0.4)
Icefall	0	1 (100)	0	0	1 (0.1)
Unknown cause	7 (70)	1 (10)	2 (20)	0	10 (0.5)
Total	960 (57.0)	489 (29.0)	62 (3.7)	173 (10.3)	1684 (100.0)

Note: Unknown: Police officers were unable to obtain information.

**Table 3 ijerph-17-00103-t003:** Difficulty level of via ferratas in relation to injury severity in Austria from 1 November 2008 to 31 October 2018. Grading was performed using the Austrian Scale by Kurt Schall (Appendix A) [4].

Difficulty Level of Via Ferrata	Uninjured *n* (%)	Injured *n* (%)	Dead *n* (%)	Unknown *n* (%)	Total *n* (%)
Grade A	38 (50.0)	22 (28.9)	10 (13.2)	6 (7.9)	76 (4.5)
Grade B	104 (46.6)	76 (34.2)	11 (4.9)	32 (14.3)	223 (13.2)
Grade C	218 (56.6)	126 (32.7)	3 (0.8)	38 (9.9)	385 (22.9)
Grade D	255 (59.9)	108 (25.3)	19 (4.5)	44 (10.3)	426 (25.3)
Grade E	74 (62.7)	40 (34.0)	1 (0.8)	3 (2.5)	118 (7.0)
Other	271 (59.4)	117 (25.6)	18 (4.0)	50 (11.0)	456 (27.1)
Total	960 (57.0)	489 (29.0)	62 (3.7)	173 (10.3)	1684 (100.0)

Note: Other was terrain of either undefined or unknown severity grading.

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
