# Peer review of "Mortality in Via Ferrata Emergencies in Austria from 2008 to 2018"

_ijerph, 2019, doi:10.3390/ijerph17010103_

Round 1
Reviewer 1 Report
The very first paragraph needs to give a detailed explanation of a via ferrata, including examples, likely participants, and how they are graded.
A summary of the facts and conclusions from the two Yosemite studies should be included, and a discussion of any differences with this study would be useful.
Figure 1 in the appendix should go in the main text with a breakdown of % cause in each year. Some comment as to why 2011 and 2014 are relatively high may be helpful.
Are there any recommendations as to how the number of rescues can be reduced, i.e. how can instances of human stupidity be even partly overcome.
Are there any other novel pieces of information that the authors can extract from the data, i.e. which age is most prone to not using safety equipment, etc.
Author Response
Reviewer 1:
The very first paragraph needs to give a detailed explanation of a via ferrata, including examples, likely participants, and how they are graded.
Thank you very much for the helpful input. In order to introduce the topic more properly, we implemented some general information on via ferratas as well and restructured the introduction. The first paragraph now starts as followed (lines #39-47):
“A via ferrata – literally translated into “iron path” – is a mountainous climbing route fully and permanently equipped with facilities like ladders, steps, cables and anchors fixed to the rock. Difficulty levels reach from easy access with many steps and footholds and no athletic passages to access difficulties (e.g. climbing sequences) and exposed vertical or overhanging sections equipped only with cables. Therefore, via ferrata climbing is practiced by a broad cross-section of the population with various experience levels and also age ranges from very old to very young. In addition to general protective gear – especially gloves, partially containing shock absorbers – protective equipment mainly consists of a harness, a helmet and a special via ferrata, heavy-duty lanyard as a connection between the harness and the steel cable in the wall. Via ferrata kits are commercially available.”
A summary of the facts and conclusions from the two Yosemite studies should be included, and a discussion of any differences with this study would be useful.
Dear reviewer, this has also been suggested by another reviewer. We fully agree with your plea.
Richardson et al. analysed 31 deaths in 85 years. There were 11 climbers exiting half dome (36%), 8 suicides (26%), 5 cable handrail-related falls (16%), 5 hikers (16%), and 2 base jumpers (6%). Weather and training status were often responsible. Morbidity and mortality impact should be performed.
Spano et al. compared in 2019 the rates of mortality and injury before and after implementing a permit to access Half dome. There was no significant reduction in mortality, concluding that crowding of via-ferrata was not the main cause. Costs of rescue are calculated. Permit might lead to stress in proceeding via-ferrata, even when exhausted as these occurred relatively frequent.
We added the following sentences:
“To our knowledge only two studies from Yosemite National Park, CA have been published to date [1,2]. Both concentrate on Half Dome, a summit reached by a 23 to 26 km round-trip hike followed by the final 146 m which are equipped with cable handrails.” (lines # 76-77)
“This is in line with data from the Yosemite National Park where an age range from 8 – 70 yrs was observed [2]” (lines # 208-209)
“On Half Dome in Yosemite, where summer weekend peaks are also observed, overcrowding has already led to restrictions in access permissions [2].” (lines #216-217)
“We observed a larger number of uninjured (57%) as compared to injured persons involved in via ferrata emergencies, which is comparable to the 59% non-traumatic search and rescue missions from Yosemite [2]. “ (lines #219-220)
“Compared to a nationwide mortality of 6.2 deaths per year in Austria the data from Yosemite show a total number of deaths of n = 12 in a ten-year timeframe for a single summit [2]. The differentiation of causes leading to death in the history of the summit revealed that 10 from 31 traceable deaths were related to hiking (n = 5) and cable handrails (n = 5) [1].” (lines #243-246)
Figure 1 in the appendix should go in the main text with a breakdown of % cause in each year. Some comment as to why 2011 and 2014 are relatively high may be helpful.
Thank you, reviewer. We decided against putting this file into the main text and added it to the supplements instead. We did follow your suggestion and broke up the years in single causes of accidents. Now we believe it is easier to see that fluctuation between years was mainly caused by exhausted persons. Both summers 2011 and 2014 were seasons of extremes, often too humid and periods of very hot. Therefore, we conclude that this might result in more persons remaining exhausted in via-ferratas.
Are there any recommendations as to how the number of rescues can be reduced, i.e. how can instances of human stupidity be even partly overcome.
Dear reviewer, although the attempt to overcome human stupidity might sound amusing, we certainly do understand the earnestness of this topic. Since via-ferratas are freely accessed there are no controls on who attempts which tour with which experience level under which conditional circumstances. In guided tours (especially with beginners or as touristic activity) we must rely on the tour guides to properly check fitness level of the participants and pick the tour wisely. Training and qualification of the tour guide should be adequate. For non-guided tours a good and even picturesque (e.g. pictured signs when entering the via ferrata, etc.) description of the difficulty level of the tour might be helpful. Standardization of difficulty scales may help as well. The UIAA decided on the Italian scale as standard for via-ferratas. Other tour guides use the Schall- or the Hülser-scale. This might lead to confusion as well as underestimation of the difficulty of the via ferrata. Unfortunately though, in the end there is nothing else for it but to rely on situation awareness and human sanity.
Are there any other novel pieces of information that the authors can extract from the data, i.e. which age is most prone to not using safety equipment, etc.
Dear reviewer, we primarily thought about presenting such data, but were afraid of overfilling the study. We must admit though, that in our opinion adding the following sentences truly improved the manuscript. Thank you!
“In the group of 20-29 year olds, females were 1.6-fold more often involved in via-ferrata accidents. In older groups men were more frequently involved (50-59 yrs 1.4-fold, 60-69 yrs 1.7-fold and in 70-79 yrs of age 2.3-fold). The mean level of altitude was 1522 ± 622 m. Injury severity was not affected by the altitude of the accident site.” (lines #108-111)
“In comparison to ground-based rescue, helicopter rescue was more frequently performed in higher altitudes (1722 ± 590 m vs. 1273 ± 645 m respectively).” (lines #113-114)
“Lastly, equipment failure or malfunction was the cause of 49 emergencies (2.9%) resulting in 17 injured patients and three fatalities. Compared to the overall population, not using proper via-ferrata equipment was 1.5-fold more frequent in the 10-19 year olds whereas the very old (70-79 yrs) were twice as often involved in accidents with equipment malfunction (6.5% vs. 2.9%). The relative risk for equipment failure was almost equal in male (3.0%) and female (2.7%), but women had a 1.9-fold higher risk for making mistakes while belaying. Incidents caused by not using a via ferrata set or equipment malfunction were 1.2- and 2.5-fold higher in men. Non-usage of a via ferrata set was also 1.8-fold higher in Austrian-natives than in the most frequent group of tourists (Germans).” (lines #135-146)
“The risk to suffer a deadly accident was increased in the old (60-69, 2.8-fold; 70-79 3.5-fold) and decreased in the young (10-19 yrs and 20-29 yrs, 0.45-fold).” (lines #157-159)
“There was a slight reduction in risk for lethal accidents when the tour was guided (1:1.3).”
(lines #172-173)
We also added the following sentence to the discussion:
“Differentiating by age and gender showed that especially young men were prone to being involved in a via ferrata accident because of non-usage of safety equipment whereas old men rather dealt with equipment malfunction. The most predominant female contribution to via ferrata emergencies were mistakes while belaying.” (lines #210-213)
Reviewer 2 Report
Dear authors,
I very much appreciate the opportunity to review your well-prepared manuscript “Mortality in via ferrata emergencies in Austria from 2008 to 2018”. I strongly agree with the authors that this is a topic of great interest as climbing via ferratas gets more and more popular and corresponding data are missing so far. I like the manuscript submitted very much, the results are very good and clearly presented and, in particular, I fully agree with the conclusions drawn in the discussion.
Therefore, I recommend to consider this manuscript for publication in Int. J. Environ. Res. Public Health.
Nevertheless, I have some minor remarks and recommendations.
First and foremost, I find the structure of the tables and figures confusing. I think it's a very good idea to offer illustrations as supplemental data for download, which are of secondary importance for the information presented in the manuscript, but for readers who are not so familiar with the subject or would like to deal with the article intensively. But I would leave it at these two categories and not add a third one as appendix. Especially since the figure A1 does not offer any essential value compared to the text, on the contrary it seems as if the interest for via ferratas has been declining since 2014. I would not include this illustration in the manuscript, but make it available for download.
Fig S1 and S2 offer the interested reader additional further information just like Tab S1 and Tab S2 (which, however, is listed as Tab 2). Tab 3 is included in the manuscript and additionally under supplemental data, the latter is unnecessary. I think Tab 3 is important and should be placed in the manuscript, but the last three lines are confusing. Even in the context of this review, I'm unsure whether the "Undefined" and "Unknown" terms refer to the via ferrata or the surrounding rocky terrain. I also doubt that the majority of readers are able to deal with the UIAA grade, especially as the difficulty levels are not given. If these three lines refer to the terrain next to the via ferrata, I would summarize them in one line.
Abstract
Line 3I-32: I don't understand the meaning of the %-indications behind the given difficulty levels for via ferratas.
Introduction
L55: References 1 and 2 are cited as the only other studies on via ferratas. It would be desirable if their data were included in the discussion.
Material and Methods
L74: “Supplemantal File 1”; I think instead of “File 1” “Fig 1” is meant
Results
Do you have data on the age distribution of accidents and deaths? As they deal with age in the discussion (L156 - 158), this information would be interesting - also in the discussion of the data.L93-99: The data of this paragraph would be more descriptive in a table.
L114: I would relate the mortality of men to that of women and not to the overall population of which men make up the majority.
L130-132: also compare my remarks on Table 3.
I consider it impossible to cope with terrain with difficulty UIAA 3 ± 1.2, i.e. up to over 4, in the context of access and exit via ferratas. Such climbing difficulties usually require rope team climbing. The vast majority of via ferrata climbers cannot cope with such difficulties, especially with via ferrata equipment. Maybe I didn't understand it correctly, but in my opinion this point needs to be explained in more detail, especially how it could happen that via ferrata climbers get into such difficult climbing terrain. Is there a need for clearer route marking??
L133: I didn't quite understand if "traversing a rock face horizontally" meant that the via ferrata was left. When the via ferrata was left, this is of course a crucial information regarding morbidity and mortality. If possible, this should be further elaborated.
Discussion
I very much like the discussion, especially the conclusions drawn from the data.
Author Response
Reviewer 2:
First and foremost, I find the structure of the tables and figures confusing. I think it's a very good idea to offer illustrations as supplemental data for download, which are of secondary importance for the information presented in the manuscript, but for readers who are not so familiar with the subject or would like to deal with the article intensively. But I would leave it at these two categories and not add a third one as appendix. Especially since the figure A1 does not offer any essential value compared to the text, on the contrary it seems as if the interest for via ferratas has been declining since 2014. I would not include this illustration in the manuscript, but make it available for download.
Thank you, reviewer. We decided to add Figure A1 to the supplements. We also broke up the years in single causes of accidents. Now we believe it is easier to see that fluctuation between years was mainly caused by exhausted persons. Both summers 2011 and 2014 were seasons of extremes, often too humid and periods of very hot. Therefore, we conclude that this might result in more persons remaining exhausted in via-ferrata.
Fig S1 and S2 offer the interested reader additional further information just like Tab S1 and Tab S2 (which, however, is listed as Tab 2). Tab 3 is included in the manuscript and additionally under supplemental data, the latter is unnecessary. I think Tab 3 is important and should be placed in the manuscript, but the last three lines are confusing. Even in the context of this review, I'm unsure whether the "Undefined" and "Unknown" terms refer to the via ferrata or the surrounding rocky terrain. I also doubt that the majority of readers are able to deal with the UIAA grade, especially as the difficulty levels are not given. If these three lines refer to the terrain next to the via ferrata, I would summarize them in one line.
Dear reviewer, thank you very much for your comments. We followed your suggestion to simplify Table 3 by summarizing the last three lines. Regarding Tab 3 and Tab S3 we must admit that they are very similar, but they are not quite alike. Tab 3 states difficulty levels vs injury severity and Tab S3 states difficulty levels vs. rescue mode.
Abstract
Line 3I-32: I don't understand the meaning of the %-indications behind the given difficulty levels for via ferratas.
Dear reviewer, the sentence questioned is: “The mortality rate was highest in technically easy to climb sections (Grade A, 13.2%/B, 4.9%), whereas the need to be rescued uninjured was highest in difficult routes (Grade D, 59.9%/E, 62.7%).”
The %-indications mean that in grade A and B via ferratas the mortality was 13.2% and 4.9% respectively (as in Tab. 3: 10 deaths from 76 emergencies in grade A and 11 deaths from 223 emergencies in grade B). The second part of the sentence refers to being rescued uninjured. Here 255 of 426 emergencies in grade D (59.9%) and 74 of 118 in grade E (62.7%) were rescued uninjured (Tab 3). Although we do admit that the %-indications referring to different injury severities can be confusing, we decided to not change the sentence as it is also summarized in Tab. 3.
Introduction
L55: References 1 and 2 are cited as the only other studies on via ferratas. It would be desirable if their data were included in the discussion.
Dear reviewer, this has also been suggested by another reviewer. We fully agree with your plea.
Richardson et al. analysed 31 deaths in 85 years. There were 11 climbers exiting half dome (36%), 8 suicides (26%), 5 cable handrail-related falls (16%), 5 hikers (16%), and 2 base jumpers (6%). Weather and training status were often responsible. Morbidity and mortality impact should be performed.
Spano et al. compared in 2019 the rates of mortality and injury before and after implementing a permit to access Half dome. There was no significant reduction in mortality, concluding that crowding of via-ferrata was not the main cause. Costs of rescue are calculated. Permit might lead to stress in proceeding via-ferrata, even when exhausted as these occurred relatively frequent.
We added the following sentences:
“To our knowledge only two studies from Yosemite National Park, CA have been published to date [1,2]. Both concentrate on Half Dome, a summit reached by a 23 to 26 km round-trip hike followed by the final 146 m which are equipped with cable handrails.” (lines # 76-77)
“This is in line with data from the Yosemite National Park where an age range from 8 – 70 yrs was observed [2]” (lines # 208-209)
“On Half Dome in Yosemite, where summer weekend peaks are also observed, overcrowding has already led to restrictions in access permissions [2].” (lines #216-217)
“We observed a larger number of uninjured (57%) as compared to injured persons involved in via ferrata emergencies, which is comparable to the 59% non-traumatic search and rescue missions from Yosemite [2]. “ (lines #219-220)
“Compared to a nationwide mortality of 6.2 deaths per year in Austria the data from Yosemite show a total number of deaths of n = 12 in a ten-year timeframe for a single summit [2]. The differentiation of causes leading to death in the history of the summit revealed that 10 from 31 traceable deaths were related to hiking (n = 5) and cable handrails (n = 5) [1].” (lines #243-246)
Material and Methods
L74: “Supplemantal File 1”; I think instead of “File 1” “Fig 1” is meant
Thank you once more for your attentive reading! We changed this embarrassing mistake to: “Figure S2”
Results
Do you have data on the age distribution of accidents and deaths? As they deal with age in the discussion (L156 - 158), this information would be interesting - also in the discussion of the data.
Dear reviewer, we primarily thought about presenting such data, but were afraid of overfilling the study. We must admit though, that in our opinion adding the following sentences truly improved the manuscript. Thank you!
“In the group of 20-29 year olds, females were 1.6-fold more often involved in via-ferrata accidents. In older groups men were more frequently involved (50-59 yrs 1.4-fold, 60-69 yrs 1.7-fold and in 70-79 yrs of age 2.3-fold). The mean level of altitude was 1522 ± 622 m. Injury severity was not affected by the altitude of the accident site.” (lines #108-111)
“In comparison to ground-based rescue, helicopter rescue was more frequently performed in higher altitudes (1722 ± 590 m vs. 1273 ± 645 m respectively).” (lines #113-114)
“Lastly, equipment failure or malfunction was the cause of 49 emergencies (2.9%) resulting in 17 injured patients and three fatalities. Compared to the overall population, not using proper via-ferrata equipment was 1.5-fold more frequent in the 10-19 year olds whereas the very old (70-79 yrs) were twice as often involved in accidents with equipment malfunction (6.5% vs. 2.9%). The relative risk for equipment failure was almost equal in male (3.0%) and female (2.7%), but women had a 1.9-fold higher risk for making mistakes while belaying. Incidents caused by not using a via ferrata set or equipment malfunction were 1.2- and 2.5-fold higher in men. Non-usage of a via ferrata set was also 1.8-fold higher in Austrian-natives than in the most frequent group of tourists (Germans).” (lines #135-146)
“The risk to suffer a deadly accident was increased in the old (60-69, 2.8-fold; 70-79 3.5-fold) and decreased in the young (10-19 yrs and 20-29 yrs, 0.45-fold).” (lines #157-159)
“There was a slight reduction in risk for lethal accidents when the tour was guided (1:1.3).”
(lines #172-173)
We also added the following sentence to the discussion:
“Differentiating by age and gender showed that especially young men were prone to being involved in a via ferrata accident because of non-usage of safety equipment whereas old men rather dealt with equipment malfunction. The most predominant female contribution to via ferrata emergencies were mistakes while belaying.” (lines #210-213)
L93-99: The data of this paragraph would be more descriptive in a table.
Dear reviewer, we fully agree. Table 2 was not indicated in the text but directly followed the chapter. In Table 2 the causes of emergencies are listed by total frequency and subdivided by injury classification. We added the reference (Table 2) after the first sentence of the paragraph.
L114: I would relate the mortality of men to that of women and not to the overall population of which men make up the majority.
Thank you reviewer. We calculated the relative risk of men and women in comparison and changed the sentence as follows:
The risk for dying was 2.5-fold higher for males than for females. (See line #157)
L130-132: also compare my remarks on Table 3.
I consider it impossible to cope with terrain with difficulty UIAA 3 ± 1.2, i.e. up to over 4, in the context of access and exit via ferratas. Such climbing difficulties usually require rope team climbing. The vast majority of via ferrata climbers cannot cope with such difficulties, especially with via ferrata equipment. Maybe I didn't understand it correctly, but in my opinion this point needs to be explained in more detail, especially how it could happen that via ferrata climbers get into such difficult climbing terrain. Is there a need for clearer route marking??
Dear Reviewer, what we could read out of written police reports, was that these persons frequently either left the route at an unforeseen location or entered it at such a position. Many of the victims tried to exit the route via such a difficult terrain. Nevertheless, we simplified this part in Table 3.
We added: …was 3.14 ± 1.2 indicating a terrain normally in need for climbing experience. (see line #181)
L133: I didn't quite understand if "traversing a rock face horizontally" meant that the via ferrata was left. When the via ferrata was left, this is of course a crucial information regarding morbidity and mortality. If possible, this should be further elaborated.
Dear reviewer, in this case "traversing a rock face horizontally” does not necessarily mean leaving the via ferrata. We rather believe that these cases indicate smaller easier sections of a via ferrata that are easily underestimated. The alpine police graded these sections according to the via ferrata grade and not according to the UIAA rock climbing scale. There are no details indicating that the emergencies took place outside the marked via ferrata route.
Reviewer 3 Report
The authors provide novel data on the mortality in via ferrata emergencies occurred in Austrian Alps for a ten year period. As such an extensive analysis in the field of via ferratas is missing up to now, this study may be of high relevance for the reader of IJERPH.
Constructive comments to the manuscript:
L 57 Materials and Method section:
Please could you indicate the search query in the method section (“113.184 did not correspond to search query”)
L 76 Statistics: please indicate details to the statistical analyses performed in this study
L 77 Result section:
The study would benefit from subgroup analysis to evaluate potential differences between male and female patients or different age groups (e.g. using simple t-tests) What is the level of altitude were the accidents took place. Is there any influence of the altitude level? Is there a lower risk for an accident when tours are guided? Could you please provide some information according to this issue? How is the proportion of Austrian and Non-Austrian patients (tourists may represent lower exercise levels/ be less accustomed to via ferratas than Austrians)L 142 Discussion:
L 162: Could you provide some data (number of uninjured to injured persons) for “This has also been described for other mountain activities”?
L 205 Limitation section
May there also be an influence of the duration of the via ferrata climbing/ fitness level of the subjects?Author Response
Reviewer 3:
Constructive comments to the manuscript:
L 57 Materials and Method section:
Please could you indicate the search query in the method section (“113.184 did not correspond to search query”)
Dear reviewer, we gladly added the following sentence to the method section:
“Cross-country skiing, skiing, ski touring, snowboarding, toboggan, canyoning, rafting, hiking, caving, mountain biking, paragliding and hang-gliding were the activities which were excluded. In a next step climbing activity was excluded when ice climbing, sport climbing, bouldering or alpine climbing was performed.” (see line #87-90)
L 76 Statistics: please indicate details to the statistical analyses performed in this study
Dear reviewer, we entered following sentence to clarify analyses:
“Differences in characteristics between victims were calculated using Chi-square tests. Descriptive statistics are presented as mean ± SD or count and percentage, as appropriate.” (see line #100-101)
L 77 Result section:
The study would benefit from subgroup analysis to evaluate potential differences between male and female patients or different age groups (e.g. using simple t-tests) What is the level of altitude were the accidents took place. Is there any influence of the altitude level? Is there a lower risk for an accident when tours are guided? Could you please provide some information according to this issue? How is the proportion of Austrian and Non-Austrian patients (tourists may represent lower exercise levels/ be less accustomed to via ferratas than Austrians)
Dear reviewer, we primarily thought about presenting such data, but were afraid of overfilling the study. We must admit though, that in our opinion adding the following sentences truly improved the manuscript. Thank you!
“In the group of 20-29 year olds, females were 1.6-fold more often involved in via-ferrata accidents. In older groups men were more frequently involved (50-59 yrs 1.4-fold, 60-69 yrs 1.7-fold and in 70-79 yrs of age 2.3-fold). The mean level of altitude was 1522 ± 622 m. Injury severity was not affected by the altitude of the accident site.” (lines #108-111)
“In comparison to ground-based rescue, helicopter rescue was more frequently performed in higher altitudes (1722 ± 590 m vs. 1273 ± 645 m respectively).” (lines #113-114)
“Lastly, equipment failure or malfunction was the cause of 49 emergencies (2.9%) resulting in 17 injured patients and three fatalities. Compared to the overall population, not using proper via-ferrata equipment was 1.5-fold more frequent in the 10-19 year olds whereas the very old (70-79 yrs) were twice as often involved in accidents with equipment malfunction (6.5% vs. 2.9%). The relative risk for equipment failure was almost equal in male (3.0%) and female (2.7%), but women had a 1.9-fold higher risk for making mistakes while belaying. Incidents caused by not using a via ferrata set or equipment malfunction were 1.2- and 2.5-fold higher in men. Non-usage of a via ferrata set was also 1.8-fold higher in Austrian-natives than in the most frequent group of tourists (Germans).” (lines #135-146)
“The risk to suffer a deadly accident was increased in the old (60-69, 2.8-fold; 70-79 3.5-fold) and decreased in the young (10-19 yrs and 20-29 yrs, 0.45-fold).” (lines #157-159)
“There was a slight reduction in risk for lethal accidents when the tour was guided (1:1.3).”
(lines #172-173)
We also added the following sentence to the discussion:
“Differentiating by age and gender showed that especially young men were prone to being involved in a via ferrata accident because of non-usage of safety equipment whereas old men rather dealt with equipment malfunction. The most predominant female contribution to via ferrata emergencies were mistakes while belaying.” (lines #210-213)
L 142 Discussion:
L 162: Could you provide some data (number of uninjured to injured persons) for “This has also been described for other mountain activities”?
Dear reviewer, we were gladly able to add some data to the mentioned sentence. Following changes have been made:
“This has also been described for other mountain activities. A nationwide retrospective study also conducted in the Austrian Alps showed 35.9% of all people involved in a canyoning related accident to be uninjured [ijerph-649020, currently under review].” (see line #220-222)
To add more information on other mountain activities we also added the following sentence:
This stands in comparison to an average annual death rate of 110 while hiking, 20 while rock climbing, 5 while mountain biking and 0.7 while canyoning in the Austrian Alps [7, ijerph-649020, currently under review].
L 205 Limitation section
May there also be an influence of the duration of the via ferrata climbing/ fitness level of the subjects?
Thank you, reviewer, for this question. We strongly believe that this is a fact. As we don’t know the fitness of the persons involved, we decided also not to evaluate the length of the via-ferrata. The police does not support data on how long the victim was already in the route.
Round 2
Reviewer 1 Report
The changes made have made the manuscript more readable and informative